# Nanocomposite Anion Exchange Membranes with a Conductive Semi-Interpenetrating Silica Network

**DOI:** 10.3390/membranes11040260

**Published:** 2021-04-04

**Authors:** Emanuela Sgreccia, Maria Luisa Di Vona, Simonetta Antonaroli, Gianfranco Ercolani, Marco Sette, Luca Pasquini, Philippe Knauth

**Affiliations:** 1International Laboratory: Ionomer Materials for Energy, Department of Industrial Engineering, University of Rome Tor Vergata, 00133 Roma, Italy; divona@uniroma2.it; 2Department of Chemical Sciences and Technologies, University of Rome Tor Vergata, 00133 Roma, Italy; simonetta.antonaroli@uniroma2.it (S.A.); ercolani@uniroma2.it (G.E.); sette@uniroma2.it (M.S.); 3Sorbonne Paris Cité, CSPBAT Laboratory, University of Paris 13, UMR 7244, CNRS, 93000 Bobigny, France; 4MADIREL (UMR 7246) and International Laboratory, Ionomer Materials for Energy, Campus St Jérôme, Aix Marseille University, CNRS, 13013 Marseille, France; luca.pasquini@univ-amu.fr (L.P.); philippe.knauth@univ-amu.fr (P.K.)

**Keywords:** polysulfone, sol–gel, ionomers, hybrid materials

## Abstract

Nanocomposite anion exchange membranes were synthesized based on poly(sulfone trimethylammonium) chloride. A hybrid semi-interpenetrating silica network containing a large amount of quaternary ammonium groups was prepared by two sol–gel routes, in situ with a single precursor, *N*-trimethoxysilylpropyl-*N*,*N*,*N*-trimethylammonium chloride (TMSP), or ex situ mixing two precursors, TMSP and 3-(2-aminoethylamino)propyldimethoxy-methylsilane (AEAPS). The properties of these hybrid composites and their degradation after immersion in 1 M KOH at 60 °C were studied. The degradation is reduced in the composite materials with a lower decrease in the ion exchange capacity. FTIR spectra showed that a main degradation mechanism with a single precursor TMSP is the dissolution of the hybrid silica network in KOH, whereas it is stable with the mixture of TMSP/AEASP. This conclusion is in agreement with the thermogravimetric analysis. The mechanical properties show a better ductility with a single precursor and higher stiffness and strength, but less ductility, by the ex situ route. The activation energy was between 0.25 and 0.14 eV for Cl and OH ion conduction, respectively, consistent with the migration mechanism.

## 1. Introduction

Anion exchange membranes (AEM) are extensively investigated for their well-known advantages when used in electrochemical devices, including the absence of noble metal catalysts for the oxygen reduction reaction in fuel cells [1,2,3,4,5,6]. However, the drawbacks of these ionomeric materials limit their extensive use and the replacement of proton exchange membranes in commercial devices [7,8]. The main AEM weaknesses are the degradation of the ionomeric groups, essentially ammonium moieties, and the scission of the backbone induced by the polarization of the matrix [9,10,11,12,13].

One of the most promising approaches to improve the properties of ion exchange polymers is the formation of organic-inorganic composite membranes [14,15,16,17,18,19]. These hybrid materials show a consistent stabilization due to the interactions between the polymer backbone and the inorganic component, including van der Waals forces and ionic interactions, e.g., between quaternary ammonium groups in the polymer or in the organosilica part with sulfone groups or ether oxygens. An extensive inorganic–organic network, possessing a structural flexibility, offers many possibilities to interact with the hydrophilic and hydrophobic part of the matrix, making it possible to change the membrane properties, especially in terms of stability [20,21,22].

However, composite ionomers suffer from a reduced conductivity due to the presence of a non-conducting second phase; this obstacle can be overcome using an intrinsically ion conducting inorganic–organic network [15,23]. Inhomogeneities that arise from the low affinity between the two phases also affect the performances of the material, leading to mechanical degradation. 

The sol–gel method is a versatile technique to produce hybrid materials and appears to be the technique of choice for the synthesis of composite membranes [24,25,26,27]. One of the strong points of the sol–gel method is its flexibility that allows the use of a large range of precursors, such as inorganic molecules (alkoxides) or hybrid molecules (ormosils). The latter can be functionalized in various ways, for example, with ionic groups. Furthermore, the second phase preparation via the sol–gel method can be modulated with the aim of obtaining materials with different morphologies, for example, by applying the sol–gel technique in-situ to prepare the silica network in the presence of the polymer or ex-situ to introduce a preformed network in the matrix. 

In this study, we choose polysulfone (PSU) functionalized with trimethylamine (TMA) as an ionomer considering its very good miscibility properties and its ability to form well-defined hydrophilic and hydrophobic domains [28]. As a second phase, we selected a hybrid silica network bearing ionic groups realized with a single precursor, such as *N*-trimethoxysilylpropyl-*N*,*N*,*N*-trimethylammonium chloride (TMSP) or by mixing two precursors such as TMSP and 3-(2-aminoethylamino)propyldimethoxy-methylsilane (AEAPS). Two sol–gel approaches were used: in situ and ex situ. The novelty of this synthetic approach is the formation of an organosilica network with an elongated or globular shape and containing ionic groups that can contribute to the ionic conductivity. The nanocomposites were analysed in terms of hydration, thermal stability, mechanical properties, ionic conductivity and degradation in alkaline conditions.

## 2. Experimental

### 2.1. Materials

Polysulfone (PSU, MW = 55,500 g/mol) was purchased from Solvay , Brussels, Belgium, and *N*-trimethoxysilylpropyl-*N*,*N*,*N*-trimethylammonium chloride from Gelest Frankfurt, Germany (50% in methanol, TMSP). 3-(2-aminoethylamino)propyldimethoxy-methylsilane (≥95%, AEAPS), trimethylamine (TMA, 4.2 M in ethanol) and all other chemicals were used as received from Sigma Aldrich St. Louis, MO, USA.

### 2.2. Synthesis of PSU-TMA

PSU-TMA was synthesized by quaternization of chloromethylated PSU (CMPSU) as described in Refs. [29,30]. 

^1^H NMR (CDCl_3_: PSU-CH_2_-Cl): δ = 1.7 (CH_3_-PSU, 6H), δ = 4.5 (PSU-CH_2_Cl, 2H), δ = 7.0–8.2 (PSU aromatic region). At δ = 7.9, hydrogens in *ortho* to the sulfone group were observed (4H). The degree of chloromethylation (DCM) was obtained by the ratio between the areas of hydrogens at 7.9 ppm, unchanged during the reaction, and hydrogens at 4.5 ppm. 

Briefly, CMPSU with degree of chloromethylation (DCM) = 0.8 was converted into a trimethylammonium salt via Menshutkin reaction using a ratio CMPSU:TMA of 1:2. The DMSO solution was stirred for 3 days at 70 °C and then heated under vacuum for 3 h at 85 °C to eliminate the excess of TMA. The final solution of PSU-TMA in DMSO (0.05 M) was directly used for the other reactions. A small amount was dried and analyzed by ^1^H NMR using DMSO-d_6_ as solvent. The ion exchange capacity (IEC) was measured by Mohr titration [31] and confirmed by NMR: IEC = 1.19 meq/g.

^1^H NMR (DMSO-d_6_: PSU-CH_2_-N(CH_3_)_3_^+^Cl^-^): δ = 1.6 (CH_3_-PSU, 6H), δ = 3.1 (N(CH_3_)_3_, 9H), δ = 4.6 (PSU-CH_2_-N(CH_3_)_3_^+^, 2H), δ = 6.75–8.25 (PSU aromatic region). The degree of amination (DAM) was measured by comparison between of the area of PSU-CH_2_-NR_3_^+^ groups and PSU methyl groups and was DAM = 0.58, corresponding to an IEC = 1.19 meq/g [32].

The DAM indicates an incomplete amination reaction with about 75% conversion rate, possibly due to second-order kinetics [5].

### 2.3. Composite Membranes

#### 2.3.1. Route 1. In Situ Sol–Gel (Synthesis of PSU-TMA/TMSP)

The sol–gel reaction was carried out in acid media, carefully adding, under stirring, 0.8 mL of 2 M HCl (1.6 × 10^−3^ mol) to 8 mL of PSU-TMA solution (0.05 M in DMSO, 4.0 × 10^−4^ mol) at 50–60 °C. Then, TMSP was added (0.13 mL, 2.4 × 10^−4^ mol) and the solution stirred for 1h at 50 °C. No precipitation was observed. The solution was transferred into a Petri dish and heated at 90 °C for 12 h in an oven. The molar ratio PSU-TMA:HCl:TMSP was 1:4:0.6.

#### 2.3.2. Route 2. Ex Situ Sol–Gel (Synthesis of PSU-TMA/TMSP-AEAPS)

Formation of the silica network. AEAPS (0.14 mL, 6.0 × 10^−4^ mol) was added to TMSP (1 mL, 1.8 × 10^−3^ mol) at RT and the solution was stirred for about 20 min. The ratio TMSP:AEAPS was 3:1. A total of 1.35 mL of 2 M HCl (2.7 × 10^−3^ mol) was then added and the pale-yellow solution heated at reflux for 24 h. After this time, the solvent was evaporated and the solid dissolved in 3.5 mL of DMSO. 

Formation of the composite membrane. A quantity of 0.12 mL of the previous solution, containing 6.0 × 10^−5^ mol of TMSP and 2.0 × 10^−5^ mol of AEAPS was added to 8 mL of PSU-TMA solution (0.05 M in DMSO, 4.0 × 10^−4^ mol) and heated at 60 °C for 30 min. The transparent and uniform solution was transferred into a Petri dish and heated at 90 °C for 12 h in an oven. The molar ratio PSU-TMA:TMSP:AEAPS was 1:0.15:0.05. 

### 2.4. Characterization

#### 2.4.1. Ion Exchange Capacity (IEC) 

After the membrane formation, the samples were intensively washed in water at 40 °C to remove possibly unreacted TMSP. The IEC was measured by Mohr titration as described in [32]. For the Mohr titration, membranes in chloride form were obtained by ion exchange in 1 M NaCl solution during 24 h. The membranes were then washed carefully with water to remove any excess NaCl. The dry membrane (after 48 h over P_2_O_5_) was then weighed and the Cl^−^ ions contained in the membranes were exchanged with SO_4_^2−^ by immersion in 1 M Na_2_SO_4_ solution during 24 h.

#### 2.4.2. NMR and FTIR Spectroscopies

^1^H NMR spectra were collected with a Bruker Avance 400 spectrometer (Billerica, Massachusetts, US) operating at 400.13 MHz using DMSO-d_6_ or CDCl_3_ as deuterated solvents. Chemical shifts (ppm) were referenced to tetramethylsilane (TMS). 

FTIR spectra were recorded between 4000 and 450 cm^−1^ using a Perkin Elmer Spectrum 2 IR spectrometer (Waltham, Massachusetts, United States) equipped with an ATR crystal diamond module. The membrane in Cl form was directly analysed by squeezing it on the diamond window at room temperature and humidity.

#### 2.4.3. Thermogravimetric Analysis (TGA) 

High-resolution TGA was performed using a TA Q500 apparatus with Pt sample holders. The experiments were made under air flux between 30 and 700 °C. The maximum heating rate was 3 K/min.

#### 2.4.4. Tensile Stress–Strain Measurements

The mechanical analysis was made on Cl form samples in an Adamel Lhomargy TESTOMETRIC M250-2.5CT testing machine (Testometric, Rochdale, UK) at room temperature and ambient humidity ((50 ± 10)% RH). The constant traction rate was 5 mm/min. Two samples of 5 mm width and 25 mm length were cut in central parts of the membranes in order to remove inhomogeneities or defects at the edges.

#### 2.4.5. Ion Conductivity

The membranes in Cl form or in OH form were obtained by treatment in 2 M NaCl or 2 M NaOH solution during 24 h at room temperature, then washed for 24 h in distilled water under nitrogen in order to limit, as much as possible, the formation of HCO_3_^−^ ions by reaction with atmospheric CO_2_. [33] The instrument employed was a VSP-300 (Biologic science instruments, Biologic, Seyssinet-Pariset, FRANCE) in PEIS mode (Potentiostatic Electrochemical Impedance Spectroscopy). The analysis was performed between 1 Hz and 6 MHz with a signal amplitude of 20 mV inside a hermetically closed Swagelok cell with stainless steel electrodes. The resistance of the membranes at 25, 45, 60 and 80 °C was obtained from a Nyquist plot by non-linear least-square fitting using a Randles equivalent circuit consisting of a series arrangement of a resistance and a resistance-constant phase element in parallel. The ionic conductivity was calculated using the resistance *R*, the membrane thickness d, typically 50–70 µm, and the electrode area A = 0.264 cm^2^:(1)$$\sigma =\frac{d}{A\times R}$$

#### 2.4.6. Water Uptake 

Before the measurement, the samples were intensively washed in water at 40 °C to remove possibly unreacted TMSP. The membranes in Cl^-^ form were dried over P_2_O_5_ during 48 h and the mass of the dry samples was measured (*m_dry_*). The samples were then hydrated in deionized water during 48 h and the mass of fully humidified membranes was obtained (*m_we_*_t_). The water uptake (in %) was calculated from the equation:(2)$$WU=100\times \frac{m_{wet}-m_{dry}}{m_{dry}}$$

#### 2.4.7. Stability Tests

Membrane samples were immersed under nitrogen in 1 M KOH solution at 60 °C. After 24 or 48 h, they were washed many times with ultrapure water to remove the excess KOH. The membranes were then treated with 1M NaCl for 24 h to measure the IEC by Mohr titration and the ionic conductivity. FTIR spectra and TGA analysis were also recorded after the degradation.

## 3. Results and Discussion

Figure 1 shows the preparation routes for the two composites. Both sol–gel syntheses were performed in acidic conditions, where the hydrolysis reaction is favoured, to obtain chain-like structures [34]. Indeed, the introduction of TSMP alone gives more extended structures, whereas a globular morphology is experienced after the addition of AEASP.

In the in-situ route, only a part of TMSP reacted, as observed by titration (see below). A full reaction could not be achieved. The reactivity of organically modified alkoxides is related to the steric hindrance of the organic substituent groups; the bulky propylammonium moiety most likely decreases the rate of hydrolysis. Furthermore, the permanent positive charge of TMSP can interact with the oxygens of the alkoxide groups stabilizing the precursor. For these reasons, the amount of inorganic network former was significantly reduced in the ex situ route and AEAPS was used to facilitate the network formation. 

The IEC were: PSU-TMA 1.19 meq/g; PSU-TMA/TMSP 1.35 meq/g; PSU-TMA/TMSP+AEAPS 1.42 meq/g. The following calculations are based on the equivalent mass of pristine PSU-TMA and the molar mass of TMSP. Given the fact that TMSP contains one ammonium group per molecule, the IEC measured for PSU-TMA/TMSP is consistent with a molar ratio of extra-ammonium groups deriving from TMSP of 0.2, less than the initial ratio of 0.6, indicating a low reactivity and a loss of TMSP during washing. For PSU-TMA/TMSP + AEASP, the IEC is consistent with the initial ratio of reactants, indicating that the reaction is complete. The water uptake is much higher for the hybrid network made from TMSP alone (WU (25 °C) = 48%, WU (60 °C) = 77%) than for TMSP + AEASP (WU (25 °C) = 9%, WU (60 °C) = 18%), showing a much higher hydrophilicity of the former.

Figure 2 shows the dependence of the IEC on the time of alkaline treatment. The IEC decreases due to the loss of some quaternary ammonium groups. The Hofmann elimination reaction is not observed here, due to the absence of a hydrogen atom in β-position to the quaternary ammonium group. The remaining IEC after 48 h of alkaline degradation is 69% of the initial value for PSU-TMA/TMSP. The membrane containing AEAPS might present a slightly enhanced alkaline stability. The titration of PSU-TMA becomes impossible after 48 h due to brittleness of the membrane showing a strong degradation of the polymer. The increased brittleness indicates a backbone degradation, as expected for polymers containing aromatic ether bonds. The most important function of the hybrid network seems to prevent the brittleness by reduction of chain scissions. This improvement might be attributed either to preferential interactions and/or the reaction of OH^−^ with the hybrid network, which itself contains ammonium groups, or to interactions between the organosilica part and the polymer.

Figure 3 shows typical tensile stress–strain curves recorded for membranes with TMSP and with TMSP and AEASP. The corresponding mechanical properties are reported in Table 1.

The PSU-TMA/TMSP membrane presents a good combination of relatively high stiffness (Young modulus > 1 GPa) and strength (~30 MPa) and some ductility (elongation at break > 15%). Stiffness and strength are below pristine PSU-TMA, but the ductility is much improved, corresponding to a higher water uptake that reduces the Van der Waals interactions between chains [30]. One can immediately observe, in Figure 3, the strong variation of the mechanical behaviour after the addition of AEAPS: the membrane becomes distinctly stiffer (Young modulus > 1.5 GPa) and stronger (~45 MPa), but the ionomer is more brittle with an elongation at break below 10%. One can attribute these changes to the particular morphology of the hybrid silica part (Figure 1): schematically, the elongated nature of TMSP probably allows gliding of the macromolecular chains, reducing stiffness and enhancing ductility, whereas the globular nature of TMSP + AEASP impedes chain movements, enhancing stiffness but reducing ductility.

The thermogravimetric curves of the membranes are quite similar (Figure 4). The initial mass loss below 100 °C is due to the removal of water. In the case of PSU-TMA/TMSP, the water loss is much lower after the alkaline treatment, in accordance with a loss of hydrophilic ammonium groups. After the addition of AEAPS, the water loss is smaller, because the hydrophilicity of the membrane decreases significantly, in accordance with the water uptake data, but the alkaline degradation does not reduce the water loss to a considerable extent, indicating a better stability of the hydrophilic ammonium groups.

The double peak around 200 °C is attributable to the quaternary ammonium groups of PSU-TMA and the supplementary ammonium and amine groups of TMSP and AEAPS. After alkaline degradation, the second peak in the derivative curve is much smaller or nearly disappears for PSU-TMA/TMSP. It can, therefore, be attributed to the ammonium groups of the TMSP part, because the residue at 700 °C, related to SiO_2_, is nearly zero after the alkaline treatment (see below). This finding is in accordance with a dissolution of the hybrid network. With TMSP + AEASP, the SiO_2_ residue remains stable after the alkaline treatment, indicating that AEASP is able to impede the dissolution of the hybrid silica network.

A small shoulder at around 400 °C before the main peak can be attributed to the loss of structural methyl groups of PSU and some remaining non-reacted chloromethyl groups. After the alkaline degradation, this peak is enhanced and shifted to 380 °C; this evolution might be attributed to the formation of hydroxymethyl groups. 

The main decomposition peak slightly below 450 °C is related to the PSU main chain decomposition [35]. This temperature of decomposition is consistent with previous reports on PSU-based ionomers, but the addition of TMSP slightly increases the thermal stability (445 °C). A slightly lower decomposition temperature is observed after the addition of AEAPS (440 °C). After the alkaline treatment, the decomposition temperature is lower for both membranes, 430 °C for PSU-TMA/TMSP and 435 °C with AEAPS. This reduction can be attributed to the hydrolysis of ether linkages in the PSU backbone that reduces the average macromolecular chain length and the related thermal stability. One might conclude that the addition of AEAPS reduces the alkaline cleavage of ether bonds in PSU. 

The remaining mass at the end of the experiment, at 700 °C, can be attributed to thermally formed silicon dioxide. For PSU-TMA/TMSP, the amount (about 7.5 mass %) is in good agreement with the equivalent mass of the ionomer (M = 740 g/eq) and the molar mass of SiO_2_, taking the nominal ratio into account. After the alkaline treatment, this amount is nearly zero, indicating that the hybrid network has been dissolved in the alkaline solution. In the case of PSU-TMA/TMSP+AEAPS, the amount of SiO_2_ is lower (about 3 mass %), in good agreement with the effective ratio, but is nearly stable after the alkaline treatment, indicating that AEAPS stabilizes the hybrid network. 

Figure 5a,b show the FTIR spectra before (black curves) and after the alkaline treatment (red curves). The main PSU-TMA absorptions remain unchanged before and after the aging. The peak at 2970 cm^−1^ can be attributed to the CH_3_ groups of quaternary ammonium, the PSU backbone and the hybrid silica network. Bands around 1580 and 1490 cm^−1^ are due to the skeletal vibration of aromatic hydrocarbons. The characteristic absorption bands at 1325 and 1150 cm^−1^ are due to the asymmetric and symmetric stretching vibration of S-O bonds of PSU and the peak at 1240 cm^−1^ is assigned to the antisymmetric vibration of the ether linkage. Absorptions of the silica network occur mainly in the 1250–700 cm^−1^ region, where peaks at high frequency at 1200–1000 cm^−1^ are related to asymmetric transversal and longitudinal stretching of Si-O-Si bonds, while signals at 1000–950 cm^−1^ are ascribed to the Si-O in-plane stretching vibrations of Si-OH groups. Bands around 830 and 800 cm^−1^ are due to the Si-C stretching and to the symmetric mode of Si-O-Si, respectively. 

In both samples, one can notice a reduction in the peak related to hydroxide groups around 3400 cm^−1^, suggesting a decrease in the hydrophilicity due to the loss of some ammonium groups, especially in the case of PSU-TMA/TMSP (Figure 5a). For this sample, one notices an intensity reduction after alkaline treatment of peaks below 1290 cm^−1^, related to the hybrid silica network, including Si-bonded alkyl group C-H bending at 1290 cm^−1^, Si-O-Si asymmetric vibration at 1095 cm^−1^, Si-OH at 1010 cm^−1^, Si-O at 910 cm^−1^, Si-C stretching at 835 cm^−1^, Si-O-Si symmetric stretching at 690 cm^−1^, and Si-O rocking mode or Si-O stretching of network defects at 550 cm^−1^ [36]. This intensity reduction is consistent with the absence of SiO_2_ at the end of the TGA experiment at 700 °C, indicating that the sol–gel part was dissolved in KOH. In the sample with TMSP + AEASP, the intensity reduction is not observed, which is in agreement with TGA experiments. 

The higher conductivity of the OH form is consistent with the higher mobility of hydroxide ions (Table 2). The relation between ionic conductivity and ion mobility in polymers was studied by us [28] and others [37]. However, the relatively low conductivity of the OH form after 24 h washing in water might be attributed to carbonatation, although the solution was kept under nitrogen. In fact, after a short wash in water, the conductivity is higher, although an effect of some remaining KOH cannot be excluded. One can notice that the reported thicknesses reflect the different water uptake of the various membranes, especially the higher hydration of OH form membranes and the reduced hydration after the degradation treatment, due to loss of hydrophilic groups. The activation energy determined from the temperature dependence of the ionic conductivity amounts to 0.14 eV for the OH form, and an average of 0.18 eV for the Cl form of PSU-TMA/TMSP and 0.25 eV for PSU-TMA/TMSP+AEAPS. The slightly lower value for the OH form is related to the Grotthuss mechanism that is possible for hydroxide ion migration. The higher value for PSU-TMA/TMSP+AEAPS is attributable to the more condensed hybrid silica phase.

## 4. Conclusions

Anion exchange membranes with a semi-interpenetrating network were made from poly(sulfone trimethylammonium) chloride with a hybrid silica network synthesised from precursors containing quaternary ammonium groups. The in situ sol–gel route used a single precursor, *N*-trimethoxysilylpropyl-*N*,*N*,*N*-trimethylammonium chloride (TMSP), whereas, in the ex situ route, two precursors, (TMSP) and 3-(2-aminoethylamino)propyldimethoxy-methylsilane (AEAPS), were mixed.

The in-situ prepared nanocomposite showed a better ductility, probably due a superior homogeneity, whereas the ex-situ samples have a high stiffness and strength. The ionic conductivity of the in-situ nanocomposite was higher than that of the ex-situ sample; the activation energy of ion conduction was in agreement with the migration mechanism of Cl and OH ions. 

The degradation in alkaline solution, studied by various experimental methods, including IEC and conductivity measurements, thermogravimetry and FTIR spectroscopy, showed the dissolution of the hybrid silica network when only one precursor was used, whereas the TMSP + AEASP samples present a stable network. Both nanocomposites have an improved alkaline stability in comparison with PSU-TMA. 

## Figures and Tables

**Figure 1 membranes-11-00260-f001:**
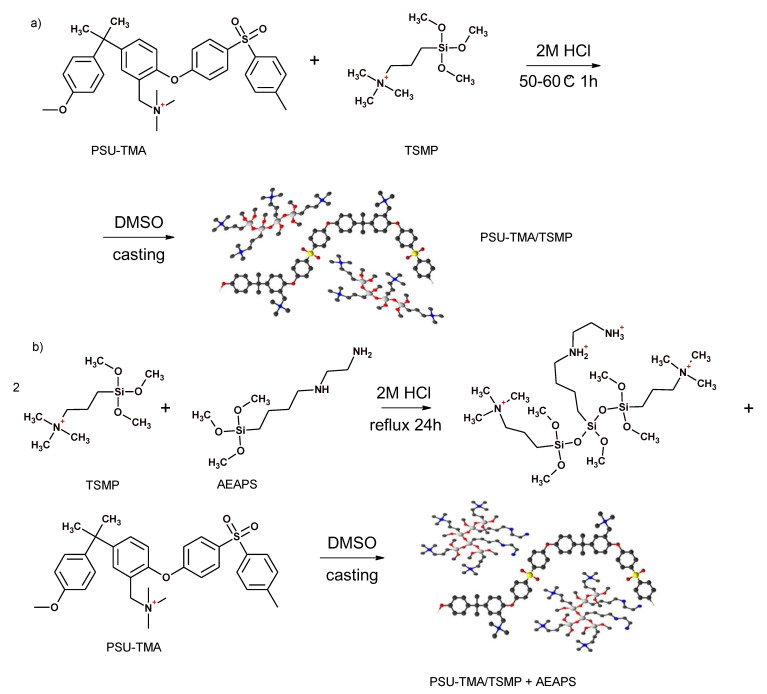
Synthesis routes for hybrid membranes by (**a**) in-situ and (**b**) ex-situ sol–gel technique.

**Figure 2 membranes-11-00260-f002:**
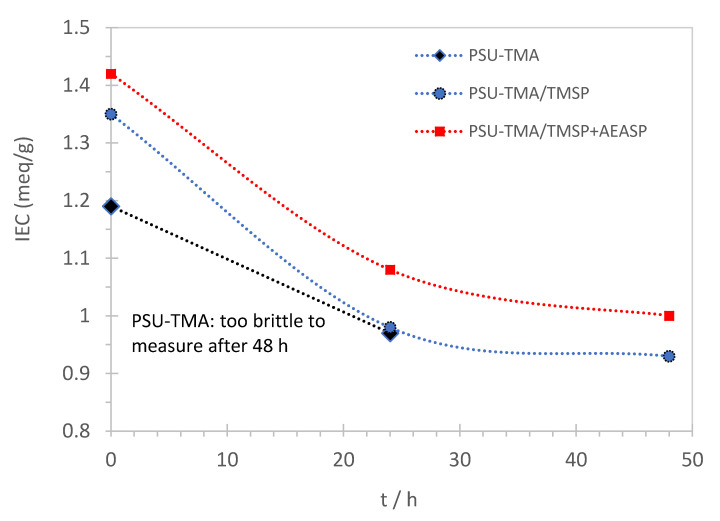
Ion exchange capacity of sol–gel membranes as function of the degradation time in KOH at 60 °C.

**Figure 3 membranes-11-00260-f003:**
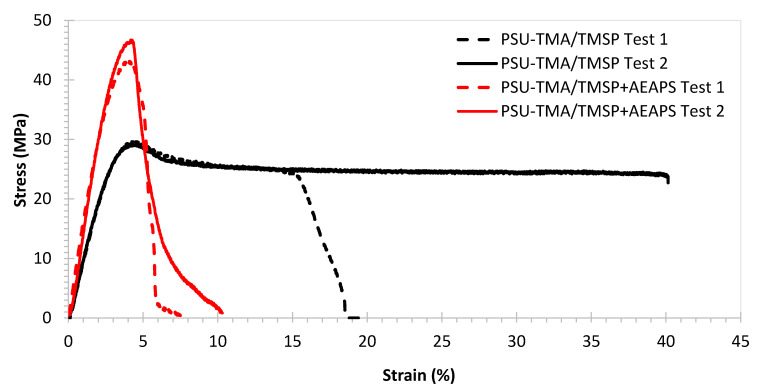
Tensile stress–strain curves for membranes in Cl^−^ form at(40 ± 2)% relative humidity and temperature ((25 ± 1) °C).

**Figure 4 membranes-11-00260-f004:**
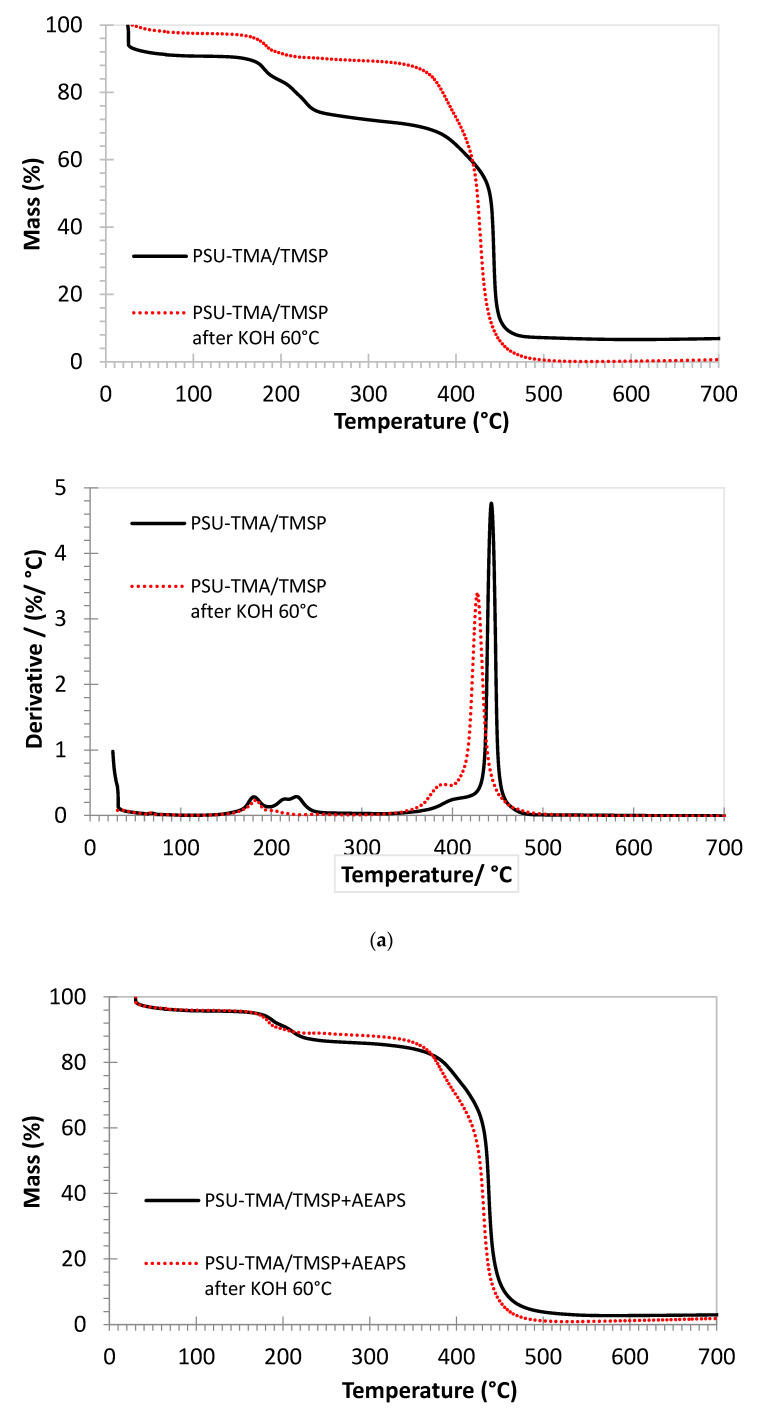
Thermogravimetric analysis of membranes in Cl form under air (mass loss and derivative curves). (**a**) PSU-TMA/TMSP, (**b**) PSU-TMA/TMSP+AEAPS. Red curves: after 24 h in KOH at 60 °C.

**Figure 5 membranes-11-00260-f005:**
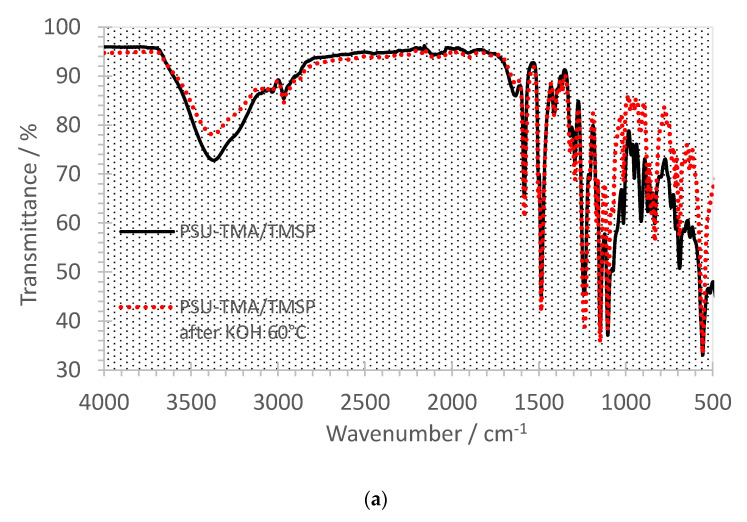
FTIR spectra of PSU-TMA membranes with silica network. (**a**) PSU-TMA/TMSP and (**b**) PSU-TMA/TMSP/AEAPS. Black curves: before treatment in KOH at 60 °C; red curves: after 24 h treatment in KOH 60 °C.

**Table 1 membranes-11-00260-t001:** Mechanical properties of membranes in Cl form at ambient humidity and temperature.

Membrane	Young’s Modulus (MPa)	Tensile Strength (MPa)	Elongation at Break (%)
PSU-TMA	1440 ± 10	44 ± 1	8 ± 1
PSU-TMA/TMSP^a^	1045 ± 13	29.5 ± 0.3	28 ± 17 *
PSU-TMA/TMSP+AEAPS^b^	1562 ± 32	45 ± 3	4.8 ± 0.4

* in the case of the dotted black curve, the sample broke slowly from one side.

**Table 2 membranes-11-00260-t002:** Ionic conductivity of membranes.

Conductivity mS/cm						
	PSU-TMA/TMSP				PSU-TMA/TMSP+AEAPS	
T/°C	OH^−^ form d = 63 µm	OH^−^ form * d = 66 µm	Cl^−^ form d = 58 µm	Cl^−^ form (after 24 h in KOH at 60 °C) d = 56 µm	Cl^−^ form d = 56 µm	Cl^−^ form (after 24 h in KOH at 60 °C) d = 50 µm
25	2.4	6.1	1.3	0.6	0.8	0.6
45	3.1	8.4	1.9	1.0	1.4	1.0
60	4.0	12.5	2.8	1.5	2.4	1.7
80	5.8	17.9	3.9	1.6	3.4	-

* after short wash in water.

## Data Availability

Not Applicable.

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
