# Peer review of "Nanocomposite Anion Exchange Membranes with a Conductive Semi-Interpenetrating Silica Network"

_membranes, 2021, doi:10.3390/membranes11040260_

Round 1
Reviewer 1 Report
In the manuscript "Nanocomposite anion exchange membranes with a conductive semi-interpenetrating silica network" authors synthesized anion exchange membranes based on poly (sulfone trimethylammonium) chloride. The thermal stability, mechanical properties, degradation, and ionic conductivity were tested. The manuscript is well written. The reviewer suggests the manuscript to be "Accepted after Minor revision". Comments: - The amount of self-citations is inappropriate (9 out of 36). Please remove some of the self-citations. - In the Figure 1 please check the conditions. Some of them differ from the method description (ex., 60 C for 1h instead of 50C for 1 h as described in methods). - In Figure 2 please change the style of the icons. They are not recognizable when printed in black/white.
Author Response
Point 1: The amount of self-citations is inappropriate (9 out of 36). Please remove some of the self-citations.
Response 1: We have now removed two self-citations and added some new citations.
Point 2: In the Figure 1 please check the conditions. Some of them differ from the method description (ex., 60 C for 1h instead of 50C for 1 h as described in methods).
Response 2: Corrected in the figure.
Point 3: In Figure 2 please change the style of the icons. They are not recognizable when printed in black/white.
Response 3: The figure was redrawn.

Reviewer 2 Report
Nanocomposite anion exchange membranes with a conductive semi-interpenetrating silica network, Emanuela Sgreccia, Maria Luisa Di Vona, Simonetta Antonaroli, Gianfranco Ercolani, Marco Sette, Luca Pasquini and Philippe Knauth
The manuscript is interesting and well-constructed. English is good. I add some observations.
Line 40.
These hybrid materials, where the interaction between the polymer backbone and the inorganic component occurs via van der Waals forces or ionic interactions, show a consistent stabilization.
Which is the interaction between the polymer and the inorganic component in this membranes? There is no interaction?
Line 83.
The ion exchange capacity (IEC) was measured by Mohr titration and confirmed by NMR: IEC = 1.19 meq/g.
The IEC for a full chloromethylated PSU would be near 2 meq/g. This partial chloromethylation was desired for any reason related with the properties of the material, or is a result of a not desired low efficiency in the chloromethylation reaction?
Line 113.
1H NMR spectra were collected with a Bruker Avance 400 spectrometer operating at 400.13 MHz using DMSO-d6 or CDCl3 as deuterated solvents. Chemical shifts (ppm) were referenced to tetramethylsilane (TMS).
Where are NMR results? They are only mentioned in IEC measurements.
Line 194.
Stiffness and strength are slightly below pristine PSU-TMA, but the ductility is much improved.
It would be better for comparison to include pristine PSU-TMA mechanical properties values in table 1, and perhaps in Figure 3.
Table 2.
It would be good to compare with the conductivity of pristine PSU-TMA, which are ca. 4 times higher (A.H.N. Rao, R.L. Thankamony, H.-J. Kim, S. Namb, T.-H. Kim, Polymer 54, 2013, 111-119). It is clear that the addition of the inorganic component improve the degradation resistance of PSU-TMA membranes, but the conductivity of the composite membrane is significantly lower than the former. It would be necessary to discuss it, and to propose ways to improve the conductivity.
Author Response
Point 1:
Line 40.
These hybrid materials, where the interaction between the polymer backbone and the inorganic component occurs via van der Waals forces or ionic interactions, show a consistent stabilization. Which is the interaction between the polymer and the inorganic component in this membranes? There is no interaction?
Response 1: We rewrote this sentence in the text to take this comment into account.
“These hybrid materials show a consistent stabilization due to the interactions between the polymer backbone and the inorganic component, including van der Waals forces and ionic interactions, e. g between quaternary ammonium groups in the polymer or in the organosilica part with sulfone groups or ether oxygens.”
Point 2:
Line 83.
The ion exchange capacity (IEC) was measured by Mohr titration and confirmed by NMR: IEC = 1.19 meq/g. The IEC for a full chloromethylated PSU would be near 2 meq/g. This partial chloromethylation was desired for any reason related with the properties of the material, or is a result of a not desired low efficiency in the chloromethylation reaction?
Response 2: In fact, a too high IEC impacts severely the hydrolytic and mechanical stability of the AEM, making them unsuitable for applications. (see e.g. Int. J. Hydrogen Energy, 39, 14039-14049 (2014)). Furthermore, the degradation is enhanced for high IEC. (see e.g. Polymer, 185, 121931 (2019)).
For this reason, we keep always a relatively low IEC.
Point 3:
Line 113.
Where are NMR results? They are only mentioned in IEC measurements.
Response 3: Now, the NMR results are detailed in the experimental part.
Point 4:
Line 194.
Stiffness and strength are slightly below pristine PSU-TMA, but the ductility is much improved. It would be better for comparison to include pristine PSU-TMA mechanical properties values in table 1, and perhaps in Figure 3.
Response 4: We have now included data for pristine PSU-TMA in Table 1. Furthermore, we have given a qualitative explanation of the various data in the text.
Point 5:
Table 2.
It would be good to compare with the conductivity of pristine PSU-TMA, which are ca. 4 times higher (A.H.N. Rao, R.L. Thankamony, H.-J. Kim, S. Namb, T.-H. Kim, Polymer 54, 2013, 111-119). It is clear that the addition of the inorganic component improves the degradation resistance of PSU-TMA membranes, but the conductivity of the composite membrane is significantly lower than the former. It would be necessary to discuss it, and to propose ways to improve the conductivity.
Response 5: We do not confirm this result. Our own data on PSU-TMA show a conductivity, which is consistent and only slightly above the one obtained with the nanocomposite. (Int. J. Hydrogen Energy, 39, 14039-14049 (2014))

Reviewer 3 Report
Minor Revision
membranes-1153322-peer-review-v1
The submitted article “membranes-1153322-peer-review-v1” reported nanocomposite anion exchange membranes which were synthesized based on poly-(sulfone tri- 16 methylammonium) chloride. The author claimed the activation energy was recorded between 0.25 and 0.14 eV for Cl and OH ion conduction which was consistent with the migration mechanism. Apart from that, however, there is points need for clear clarifications and more explanations, and thus a minor revision is required.
Comments are as follows:
Reviewer comments
- In the introduction section, the authors mentioned “The sol-gel method is a versatile technique to produce hybrid materials and appears the technique of choice for the synthesis of composite membranes”
What is a novelty in their synthesis method? explain with suitable reason or add a sentence in the synthesis section.
- As mentioned, In the in-situ route, only a part of TMSP reacted. Write the reason?
- In figure 5, the author needs to mention the bonding value of each peak.
- The higher mobility of hydroxide ion affect the conductivity? Explain and cite the appropriate article.
- The author needs to check English throughout the text, typos error, and spacing between words.
- All references must be on the pattern of the journal guidelines, consider all the commas, full stops, etc,
- Overall, the writing and included information seem reasonable except for the above-mentioned comments, the author needs to include all the suggestions before acceptance.
Author Response
Point 1: In the introduction section, the authors mentioned “The sol-gel method is a versatile technique to produce hybrid materials and appears the technique of choice for the synthesis of composite membranes”
What is a novelty in their synthesis method? explain with suitable reason or add a sentence in the synthesis section.
Response 1: We have now given novel aspects of our synthesis method. This part is now added in the introduction.
Point 2: As mentioned, In the in-situ route, only a part of TMSP reacted. Write the reason?
Response 2:We have add the discussion in the text.
“A full reaction could not be achieved. The reactivity of organically modified alkoxides is related to the steric hindrance of the organic substituent groups, the bulky prop-ylammonium moiety most likely decreases the rate of hydrolysis. Furthermore, the TMSP permanent positive charge can interact with the oxygens of the alkoxide groups stabilizing the precursor.”
Point 3: In figure 5, the author needs to mention the bonding value of each peak.
Response 3: We have added a grid in Figure 5 to facilitate the lecture of the peak values. Adding all peak values would create great confusion in the figure. All main peaks are discussed in the text.
Point 4: The higher mobility of hydroxide ion affect the conductivity? Explain and cite the appropriate article.
Response 4: According to the definition of conductivity, there is a direct relation between ionic conductivity and ion mobility. We have added a new reference on this point in the text.
Point 5: The author needs to check English throughout the text, typos error, and spacing between words.
Response 5: Done
Point 6: All references must be on the pattern of the journal guidelines, consider all the commas, full stops, etc.
Response 6: We noticed indeed some errors in the references, which are not present in our original manuscript (Pasquin, Knaut…), but have been introduced during the editing process. We checked them again.

Round 2
Reviewer 2 Report
I recommend accepting the manuscript in the present form.
Author Response
Point 1: You replied that the Figure was redrawn. However, all data points are filled circles – this looks same in grey scale, and the ranking of the curves differs from that of the legend. Please change to circle and square, etc., so that every reader can immediately catch the difference.
Response 1: Done
Point 2: it could be misleading that you put there "not measurable" - what about "PSU-TMA: too brittle to measure after 48 hours".
Response 2: Done
